# Improving Black-Box LLMs with Feedback Reinforcement Learning

## Abstract

The optimization of black-box large language models (LLMs) presents significant challenges. While the implementation of pre-existing chain-of-thought (CoT) prompting and feedback mechanisms help occasionally, these approaches still struggle from unreliable feedback, and fail to leverage the training data. In this work, we propose Feedback Reinforcement Learning (FRL)—training a separate feedback model through reinforcement learning to improve the main black-box LLM. FRL divides self-correction into two stages: our trained feedback model identifies error, and generate corresponding feedback on how to correct the error, while the black-box LLM generates correction based on this extra feedback. During training, the feedback model generates feedback rollouts for initial responses from a fixed pretrained model, which then produces revised responses. The improvement between initial and revised responses serves as the reward signal. This approach treats the solver model as a black-box and optimizes it with a separate feedback provider, enabling targeted improvement without modifying the base model. We evaluate FRL on generated Sudoku puzzles, GSM8K, and MMLU-STEM questions, demonstrating consistent improvements over the initial language model's performance by $16.5\%$ on average. Our method outperforms both non-learning self-correction approaches and oracle-based verification methods by leveraging training data through reinforcement learning. Moreover, FRL models can also function as problem solvers, outperforming their pretrained counterparts, effectively enhancing the model's original reasoning capabilities.

## 1 Introduction

Large Language models deployed as APIs present a fundamental challenge: how can we improve their performance when we cannot access or modify their weights? This black-box optimization problem is increasingly critical as organizations rely on proprietary models for customized production deployment. In this work, we propose treating these black-box models as students that can benefit from step-by-step verification and correction from a separate trainable teacher. Compared to a direct revision of the answer, the feedback has the flexibility to capture meta-level information which generalizes better across different instances of mistakes in the same category.

Our goal is to effectively improve the performance of black-box models, which is particularly important given the current ubiquity of closed-source models. Without access to the underlying model weights or the ability to either compute or apply gradient updates, direct finetuning is inapplicable in the black-box setting. In-context learning (ICL) only uses prompts, but has limited ability to improve performance. A more complex approach involves self-correction, where a model attempts to refine its initial answer. Though this approach has some promise, it is limited by the fact that models are often unable to reliable identify their own mistakes (Zhang et al., 2024; Wu et al., 2024; Huang et al., 2024; Tyen et al., 2024). Song et al. (2025) even note that models which produce better initial answers become worse at detecting their own mistakes. More effective and accurate feedback, however, could resolve this issue and allow for greater performance improvements.

To address this challenge, we present Feedback Reinforcement Learning (FRL), a method specifically designed for black-box LLM optimization. In FRL, we train a separate feedback model $F$ to identify errors in the initial responses sampled from a solver model $S$. The black-box solver model then uses this feedback to refine its answers, allowing it to improve without any weight updates. By

training $F$ to generate useful feedback, we allow it to learn how to detect and correct errors. This also allows $S$ to use incorporate the generated feedback through ICL, which has allowed pretrained LLMs to effectively refine their outputs (Kamoi et al., 2024; Shinn et al., 2023). During training, we generate an initial answer with solver model $S$, then use the feedback model $F$ to generate feedback rollouts. Conditioning on the generated feedback $f$, we use the solver model $S$ to generate a refined answer. The reward comes from whether the refinement generated using the feedback is an improvement over the initial response, as well as from the verification accuracy of the feedback model $F$.

We study the effects of FRL in two settings. First, when the models $F$ and $S$ share the same architecture, which we call same-model correction, and second when they have different architectures, which we call cross-model correction. Empirical results show that FRL improves the black-box performance by 16.5% across datasets. We note in particular that with cross-model correction, which more closely matches real-world black-box settings, we observe substantial improvements. This validates that FRL can be used in practical settings for proprietary API models.

Therefore, we make the following contributions:

1. We propose Feedback Reinforcement Learning (*FRL*), a new method to optimize black-box LLMs by training a feedback model with reinforcement learning to help it correct its mistakes.

2. We show that *FRL* achieves improvement over strong baselines across domains consisting of mathematical problems, STEM related multiple choice questions, and Sudoku puzzles.

3. We apply *FRL* to black-box LLMs from other model families and still see strong benefit, showing its practical usage to real-world proprietary models.

## 2 RELATED WORK

### 2.1 SELF-CORRECTION

Self-correction is defined as an LLM refining its answer during inference, with or without access to external knowledge. There are two main approaches to achieve this, one is by fine-tuning a model to provide feedback and correct the answer, (Kamoi et al., 2024; Madaan et al., 2023; Saunders et al., 2022; Kumar et al., 2024). This fine-tuning can be achieved either by supervised fine-tuning (Madaan et al., 2023; First et al., 2023; Saunders et al., 2022) or reinforcement learning (Bensal et al., 2025; Kumar et al., 2024; Akyürek et al., 2023). Another approach is to leverage the in-context learning ability of the LLM and guide it to correct its mistakes by prompting. (Li et al., 2024; Wang et al., 2024; Jiang et al., 2025; Madaan et al., 2023; Shinn et al., 2023) However, this approach is not very effective on reasoning tasks, and some of the settings are unrealistic with oracle answers available during test-time.

### 2.2 SELF-VERIFICATION

While self-correction focuses on both identifying and fixing errors, self-verification addresses the crucial first step, determining whether an answer is correct. This distinction matters because verification is often easier than generation (Zelikman et al., 2022). The most straightforward approach of self-verification would be to prompt the model to revise its own answer (Zhang et al., 2024; Madaan et al., 2023). However, this approach is shown to be insufficient for hard reasoning problems (Zhang et al., 2024). There are approaches that assign a confidence score to the answer (Miao et al., 2023; Taubenfeld et al., 2025; Weng et al., 2023). This approach works well when sampling multiple answers from the model. Self-verification is a crucial part of the feedback model in FRL, the feedback model needs to first verify the answer before passing it to the refinement step.

### 2.3 OPTIMIZING BLACK-BOX MODELS

A number of works have studied methods for optimizing, adapting, or tuning black-box models. In the absence of explicit gradient information, zeroth-order methods have been used to compute approximate gradients for tuning (Park et al., 2025; Hu et al., 2024; Guo et al., 2024b; Malladi et al., 2023; Zhan et al., 2024; Tsai et al., 2020). Aside from zeroth-order approximations, evolutionary algorithms can be used to optimize continuous prompts (Sun et al., 2022a;b), particularly when

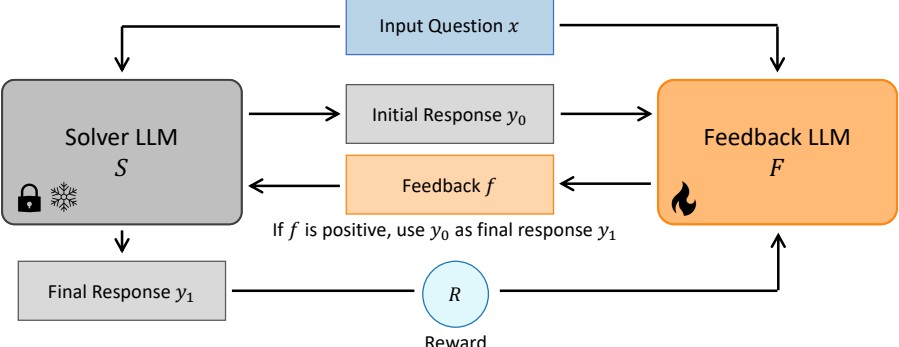

Figure 1: Our proposed FRL paradigm: Given an input question $x$, the solver generates initial output $y_0$ with solver model $S$. Then it passes question $x$ and initial response $y_0$ to the feedback model $F$ to get feedback $f$. If $f$ is positive, indicating initial response $y_0$ as correct, we use initial response $y_0$ as final response $y_1$. Otherwise, we generate final response $y_1$ with extra feedback again with solver model $S$. The reward is then computed and use to train $F$ with RL.

paired with search (Sun et al., 2024), as well as to evolve better discrete prompts (Guo et al., 2024a). Outside of evolution, reinforcement learning (Deng et al., 2022; Diao et al., 2023), search (Prasad et al., 2023), and retrieval (Cheng et al., 2023) can be used to adapt models with access to gradient information. Still others have leveraged auxiliary language models for adaptation. Ormazabal et al. (2023) use a small language model to adapt a larger one, but require access to the larger, black-box model's log probabilities. Chen et al. (2024), Zhou et al. (2023), Cheng et al. (2024), and Yang et al. (2024) each use LLMs to optimize or generate prompts for a black-box model directly. Finally, Zhou et al. (2025) use a continually update memory system to adapt a black-box model. We propose a new way of optimizing black-box LLMs by training a small feedback model to correct black-box model's mistakes.

## 3 FEEDBACK REINFORCEMENT LEARNING

Feedback Reinforcement Learning is a modular approach to correcting errors from black-box LLMs. We divide the process of self-correction into two steps: error identification and error correction. With FRL, we only train the feedback model $F$, whose job is to identify errors from the initial response $y_0$ and provide feedback to the solver model $S$ on how to correct them, so that the solver model $S$ can leverage its inherent in-context learning ability to correct the errors with extra feedback context to get the final response $y_1$.

**Inference:** During test time, given an input question $x$ and pre-trained solver model $S$, FRL generates an initial output $y_0$:

$$y_0 = S(x).$$

We prompt the model to generate this initial response using CoT, which also allows the feedback model to more easily point out specific errors. Next, we pass the input question $x$, along with initial response $y_0$ to the trained feedback model $F$ to get the feedback $f$:

$$f = F(x, y_0)$$

The feedback model analyzes each step of the initial response, identifies errors, and provides guidance on how to correct them. The feedback $f$ concludes with a verification assessment predicting the overall correctness of the initial response $y_0$. If the verification points out that the initial response $y_0$ is correct, we use it as the final response $y_1$. Otherwise, we pass the input question $x$, initial response $y_0$, and feedback $f$ to the solver model again to get the final response $y_1$:

$$y_1 = S(x, y_0, f)$$

This selective refinement makes FRL less likely to change correct initial responses to incorrect ones during refinement, and also improves efficiency by only performing refinement when needed.

**Training:** When training with FRL, we initialize the feedback model either from the same base model as the solver or from a separate model. We keep the training workflow similar to test time, where we first generate the initial response $y_0$ from $S(x)$, then using the initial response $y_0$ and input question $x$, we get the feedback rollouts $f$ from $F$. Finally, if feedback $f$ is negative, meaning the $F$ thinks the initial response is incorrect given the input question, we pass the input question $x$, initial response $y_0$, and feedback $f$ to the solver model $S$ again to get the final response $y_1$.

**Reward:** The goals for the feedback model $F$ are: to first verify the initial answer and determine its correctness and then, if the initial answer is deemed as incorrect according to $F$, to generate feedback which is helpful for the solver model to learn from. Towards these goals, we propose the following rewards:

- **Verification Reward** $R_{\mathbf{verify}}$: The feedback model receives a positive verification reward when it correctly identifies whether $y_0$ is correct or incorrect and receives a reward of $-1$ when it incorrectly identifies a correct initial response as wrong. This reward shaping is crucial because incorrectly flagging correct answers for refinement can degrade performance significantly, since initially correct solutions may become incorrect after unnecessary revision.

- **Refinement Reward** $R_{\mathbf{refine}}$: This reward assesses the quality of the feedback $f$ by measuring the improvement from the initial response $y_0$ to the final response $y_1$. To make the training as realistic as possible, we directly compare performance gains between these two responses. If the initial response $y_0$ is incorrect and final response $y_1$ is correct, $R_{\mathrm{refine}}$ is 1, 0 otherwise.

**Data Generation:** To train the feedback model effectively, we first generate training data by sampling from the solver model's outputs. For each question in the training set, we iteratively generate responses until we obtain a balanced dataset of positive examples (correct initial responses) and negative examples (incorrect initial responses). This iterative sampling approach offers two key advantages. First, it allows us to generate unlimited training examples, removing the constraint of limited question availability in the original dataset. Second, we can control the ratio of positive to negative examples to prevent training bias. Since model performance varies across datasets and model sizes (e.g., a 7B model achieves high accuracy on GSM8K, creating an imbalanced dataset), we maintain a 1:1 ratio of positive to negative examples across all experiments to ensure unbiased feedback training.

## 4    EXPERIMENTS

The goal of our experiments is to demonstrate the effectiveness of FRL on optimizing black-box LLMs through self-correction in a realistic setting. To achieve that, we chose three datasets from different domains and evaluated FRL on different model sizes, including cross-model training.

**Datasets:    Grade School Math (GSM8K)** (Cobbe et al., 2021) is a mathematical dataset consisting of 7473 training and 1319 test problems of grade-school-level word problems. These problems require multiple steps to get to the final answer. This dataset is particularly well-suited for evaluating FRL because: first, errors can occur at any intermediate step, making it crucial to identify which specific part of reasoning is wrong instead of only checking the final answer; second, the step-by-step solution allows our feedback model to provide specific, actionable feedback about the initial response, so the solver model can correct its initial mistakes with this effective feedback.

**The Massive Multitask Language Understanding (MMLU)** benchmark (Hendrycks et al., 2021) is a dataset that evaluates models across 57 subjects, including STEM, US history, law, and more. For our experiments, we focus on the STEM subset consisting of mathematics, physics, computer science, etc. These questions are all multiple-choice questions testing the conceptual understanding and multi-step reasoning of LLMs. Unlike GSM8K or Sudoku, where solutions follow explicit step-by-step procedures, MMLU-STEM problems often require implicit reasoning chains and domain-specific knowledge application. Therefore, MMLU-STEM evaluates whether FRL can generalize beyond step-by-step problem solving by requiring the feedback model to identify conceptual errors rather than just computational mistakes across diverse, less structured solutions.

**Sudoku** puzzles are a great way to test the step-by-step error identification and refinement for FRL. We generated a dataset of 4×4 Sudoku puzzles with 4 empty cells per puzzle, creating a simplified

---

**Algorithm 1** Feedback Reinforcement Learning

---

**Require:** Pretrained LLM $S$; Pretrained LLM $F$; dataset $\mathcal{D} = \{(x^i, y^i)\}_{i=1}^N$; learning rate $\alpha$
1: Sample training dataset $\mathcal{D}_{train} = \{(x^i, y^i, y_0^i)\}_{i=1}^N$ from $\mathcal{D}$ with $S$
2: **for** each $(x^i, y^i, y_0^i) \in \mathcal{D}_{train}$ **do**
3:    $f = F(x, y_0)$                                          ▷ Generate Feedback with F
4:    **if** $f$ is negative **then**
5:       generate $y_1$ from $S$ with feedback
6:    **else**
7:       $y_1 = y_0$
8:    **end if**
9:    **if** $f$ correctly identifies correct $y_0$ as correct, incorrect $y_0$ as incorrect **then**
10:      $R_{\text{verify}} = 1$                              ▷ Correct Verification
11:    **else if** $f$ identifies correct $y_0$ as incorrect **then**
12:      $R_{\text{verify}} = -1$                    ▷ False Positive Reward Shaping
13:    **else**
14:      $R_{\text{verify}} = 0$
15:    **end if**
16:    **if** $y_0 \neq y$ **and** $y_1 = y$ **then**
17:      $R_{\text{refine}} = 1$
18:    **else**
19:      $R_{\text{refine}} = 0$
20:    **end if**
21:    $R_{\text{final}} = R_{\text{refine}} + R_{\text{verify}}$
22:    $F \leftarrow F + \alpha \nabla_{\theta_f} J_{GRPO}(\theta_f)$            ▷ Update F's parameters with GRPO
23: **end for**

---

but still challenging constraint satisfaction problem. In 4×4 Sudoku, each row, column, and 2×2 sub-grid must contain the digits 1-4 exactly once. Despite this simplification from standard 9×9 Sudoku, we find that base models struggle significantly with these puzzles; even the 7B model achieves only 35.1% accuracy. This difficulty arises because Sudoku requires maintaining multiple simultaneous constraints where a single incorrect placement can make the puzzle unsolvable, making it particularly well-suited for testing whether feedback models can identify specific constraint violations and help the correction of solver models.

**Baselines:** We evaluate FRL against five baselines, which are chosen to highlight the current verification bottleneck challenge in the self-correction domain. Because we are treating the solver model as a black-box, all baselines do not require access to the model weight.

- **Sover Initial Response** establishes a lower bound for performance by showing the base model's performance with CoT prompting but without self-correction.

- **Vanilla Feedback** shares the same two-stage pipeline as FRL, but the feedback model $F$ is not finetuned, in order to demonstrate the importance of training on the feedback model $F$.

- **Self-Refine** and **Reflexion** represent state-of-the-art training-free self-correction methods that uses prompting and iterative refinement. These baselines test whether sophisticated prompting strategies can overcome the verification bottleneck.

- **Oracle Verification** uses ground truth labels to inform the solver model when its initial response is incorrect, prompting it to generate again, but without providing any specific feedback about what the error is. This baseline demonstrates that even with a perfect verifier, performing self-correction without helpful feedback is still ineffective.

- **Vanilla Feedback + Oracle Verification** combines perfect verification with the pretrained model's feedback generation. We tell the feedback model this is wrong if the initial response does not match the ground truth label, then let it generate feedback. This baseline helps us assess the upper bound of perfect verification, but untrained feedback generation.

**Experimental Protocol:** We used 1.5B, 3B, and 7B base models from the Qwen2.5 family (Qwen et al., 2025) for training. We chose base models for both feedback model $F$ and solver model $S$ to

Table 1: Performance comparison of FRL and baselines on GSM8K, MMLU-STEM, and Sudoku

| Method | GSM8K | | | MMLU-STEM | | | Sudoku | | |
|---|---|---|---|---|---|---|---|---|---|
| | 1.5B | 3B | 7B | 1.5B | 3B | 7B | 1.5B | 3B | 7B |
| Sover Initial Response | 62.7 | 76.0 | 85.4 | **54.2** | 65.1 | 69.6 | 1.7 | 4.0 | 35.1 |
| Vanilla Feedback | 62.5 | 75.8 | 85.4 | 53.9 | 63.7 | 69.7 | 1.7 | 4.0 | 34.9 |
| Self-Refine (Madaan et al., 2023) | 60.6 | 76.0 | 85.4 | 47.1 | 64.8 | 69.7 | 1.7 | 3.8 | 35.1 |
| Reflexion (Shinn et al., 2023) | 63.6 | 75.8 | 85.4 | 19.6 | 65.3 | 64.9 | 1.7 | 3.8 | 34.7 |
| Oracle Verification | 62.9 | 76.0 | 85.7 | **54.2** | 65.7 | 70.2 | **1.8** | 4.1 | 35.1 |
| Vanilla Feedback + Oracle Verification | 63.1 | 76.0 | 85.9 | **54.2** | 65.7 | 70.0 | **1.8** | 4.2 | 35.5 |
| FRL (Ours) | **75.1** | **83.1** | **91.8** | **54.2** | **69.8** | **75.8** | **1.8** | **54.0** | **96.8** |

minimize the influence of any post-training. During evaluation, we used 5-shot CoT prompting for initial solving and 2-shot CoT prompting for feedback and refinement generation due to memory and context length constraint. We prompted the models to put answers inside boxes and extract the numbers for GSM8K and MMLU-STEM. For Sudoku, we prompted the model to write a grid and extract the final grid during evaluation. We evaluated on the test set of each benchmark, ensuring no data leakage. When evaluating Self-Refine and Reflexion, we set the max number of trials to 5. For training, we first sampled the solver iteratively to generate 20,000 question-answer pairs and ran 1 epoch with a learning rate of $1 \times 10^{-6}$ and KL-penalty of 0, following (Liu et al., 2025). We used GRPO with effective batch size of 432, number of generations of 24, and temperature of 0.7. For the final reward, we combined refinement reward $R_{\text{refine}}$ and verification reward $R_{\text{verify}}$ by taking the average.

## 5 RESULTS

### 5.1 MATHEMATICAL PROBLEMS: GSM8K

In Table 1, FRL outperforms all baseline approaches by significant margins with gains of $12.4\%$, $7.1\%$, and $6.4\%$ absolute improvement for the 1.5B, 3B, and 7B models, respectively. Notably, the smaller models benefit more from our approach, with 1.5B improving from $62.7\%$ to $75.1\%$, nearly achieving the accuracy of the pretrained 3B model. This suggests that with the help of the trained feedback model, small models can also self-correct their mistakes. We analyze the failure of the prompting-based method (Self-Refine and Reflexion). For Reflexion, it shows that using a pretrained LLM to act like an evaluator is not enough, especially when the evaluator model and solver model are the same one. It tends to get overconfident and predict every answer as correct. This leads to some premature termination. Although we set max trial to 5, $96.7\%$, $96.5\%$, and $98.0\%$ of the attempts from 1.5B, 3B, and 7B stop at the first trial, meaning the evaluator predicts the initial response as correct. This essentially makes most of the response not go through the self-correction loop at all. The same thing happens to Self-Refine, where $83.3\%$, $97.7\%$, and $97.3\%$ of the attempts from 1.5B, 3B, and 7B stop at the first trial. This unreliable verification makes the two prompting-based approaches fail at this task. Evaluating on the Vanilla Feedback + Oracle Verification and Oracle Verification baseline, we see an improvement bigger than Self-Refine and Reflexion. This indicates that with more reliable verification of the initial response, a vanilla feedback model can still give helpful feedback to the solver model to correct some of its mistakes. Comparing Vanilla Feedback + Oracle Verification to Oracle Verification reveals that when pretrained models attempt to generate feedback, they can correct some errors that remain unfixed when only given binary correctness signals. However, the improvement is limited compared to FRL, and therefore training a feedback model with reinforcement learning is the critical ingredient. These findings demonstrate that the key bottleneck in self-correction is not only verification accuracy but also the ability to generate specific, actionable feedback. This ability is successfully learned by FRL through reinforcement learning.

**GSM8K Cross Model Evaluation:** Beyond same-model self-correction, we evaluate FRL's generalization capabilities in two realistic scenarios with results shown in Table 2 and Table 3. First, we test asymmetric configurations where we train smaller feedback models paired with bigger solver models. Specifically, we trained a 3B feedback model with a 7B solver model and a 7B feedback model with a 14B solver model. At test time, we evaluate whether the smaller, more efficient trained feedback model can still improve stronger, bigger solver models. Second, we simulate real-world

Table 2: Cross Family Evaluation on GSM8K

| Method | Llama | | | Deepseek | | |
|---|---|---|---|---|---|---|
| | 1.5B | 3B | 7B | 1.5B | 3B | 7B |
| Solver Only | 61.1 | 61.1 | 61.1 | 69.9 | 69.9 | 69.9 |
| Qwen Vanilla Feedback | 60.8 | 61.2 | 62.8 | 68.7 | 69.9 | 70.0 |
| Qwen FRL (Ours) | **74.3** | **77.6** | **84.4** | **73.8** | **74.8** | **78.9** |

Table 3: Asymmetric Model Training: Smaller Feedback Models with Larger Solvers on GSM8K

| Method | Feedback Model | Solver Model | Accuracy |
|---|---|---|---|
| Vanilla Feedback | 3B | 7B | 85.4 |
| Vanilla Feedback | 7B | 14B | 88.9 |
| Vanilla Feedback | No Feedback | 14B | 88.9 |
| FRL | 3B | 7B | 88.4 |
| FRL | 7B | 14B | 89.6 |

deployment by treating the solver model as a complete black-box that we do not use during training. During training, we exclusively used Qwen2.5 feedback models with Qwen solver models. Then at test time, we evaluate these Qwen-trained feedback models with solver models from completely different model families and sizes, deepseek-math-7b-base and Llama-3.1-8B. (Shao et al., 2024; Grattafiori et al., 2024) Notably, the feedback models never see or train with Deepseek or Llama during training, so they must generalize their learned feedback strategies from Qwen to these unseen structures. This cross-family evaluation demonstrates FRL's practicality and versatility, where proprietary solver models are only accessible as black-box APIs at deployment time.

The cross-family generalization results in Table 2 demonstrate FRL's remarkable ability to transfer learned feedback strategies across different model architectures. We see substantial improvements on both Llama-3.1-8B and Deepseek-Math-7B-Base solvers despite the feedback model never encountering these solver models during training. For the Llama model with a baseline of $61.1\%$, FRL feedback model provides gains of $13.2\%$, $16.5\%$, and $23.3\%$ absolute improvement for the 1.5B, 3B, and 7B feedback models, respectively. Even our smallest 1.5B model elevates Llama's performance by a great margin. Looking at the results from using Deepseek-Math-7B-Base as the solver, we also see improvement of $3.9\%$, $4.9\%$, and $8.8\%$ over its $69.9\%$ baseline. By comparing our results to the pretrained Qwen feedback models, we see only negligible or even negative improvement after the feedback, confirming that FRL training is essential to make cross-family self-correction with feedback. The consistent improvement of FRL FRL-trained model also shows that the benefit of FRL is generalizable and can provide valuable feedback to the errors that are made by Qwen models. The fact that our smallest 1.5B trained model can still effectively guide an 8B Llama model demonstrates that feedback quality matters and FRL can be used to efficiently optimize bigger black-box solver models. The cross model experiments in Table 3 demonstrate FRL's practical utility for bigger black-box LLMs. The performance of a 3B feedback model pairing with a 7B solver model improves the 3B baseline by $12.4\%$, outperforming 3B FRL by $5.3\%$. This shows that FRL can be an efficient way to train the model when pairing a small feedback model with a large solver model. This improvement also scales to a 7B feedback model pairing with a 14B solver model. This combination boosts the performance by a small margin, indicating smaller solver models benefit more when pairing with smaller feedback models.

## 5.2 MULTIPLE CHOICE REASONING: MMLU-STEM

Unlike GSM8K, where small models like 1.5B or 3B benefit the most, MMLU-STEM shows the opposite pattern in Table 1. The 7B model benefits the most from FRL, with accuracy improving from $69.6\%$ to $75.8\%$. The 1.5B model shows no improvement at all, and 3B model shows a modest gain. This pattern reveals important insights about when and how feedback-based correction succeeds. There are several factors contributing to this result. First, MMLU-STEM questions require domain-specific knowledge that smaller models may lack entirely. When a 1.5B model does not know a physics formula or biology terminology, no amount of feedback can compensate for this knowledge gap. Reinforcement cannot teach specialized knowledge that the model never learned

Table 4: Comparison to a GRPO-trained solver.

| Method | GSM8K | | | MMLU-STEM | | | GSM-Symbolic | | |
|---|---|---|---|---|---|---|---|---|---|
| | 1.5B | 3B | 7B | 1.5B | 3B | 7B | 1.5B | 3B | 7B |
| Solver GRPO | 65.1 | 76.0 | 90.8 | 53.9 | 60.4 | 78.8 | 37.2 | 60.4 | **78.2** |
| FRL (Ours) | **75.1** | **83.1** | **91.8** | **54.2** | **69.8** | **78.9** | **49.8** | **68.1** | 77.0 |

during pretraining. This limitation also shows for prompting based approach like Reflexion, where the 1.5B model achieves negative gain. Without a basic understanding of the questions and concepts, the model cannot generate meaningful verification or reflection. Doing multiple rounds of reflection would just hurt the performance more and lead to collapse. Looking more closely at the results, we see that the LLM evaluator was not able to follow the instruction and give a binary prediction, given the answer for a smaller model like 1.5B in MMLU-STEM. Because the task is beyond the capability of the small models. With the FRL-trained feedback model, we also find cases where the feedback model attempts to correct domain knowledge, but sometimes provides incorrect information. This behavior shows the limitation of FRL: while procedural errors in mathematics can be verified and corrected through step-by-step feedback, conceptual knowledge requires the feedback model to have this knowledge during pretraining.

### 5.3 Reasoning Puzzle: Sudoku

In the Sudoku dataset, we see the greatest improvement compared to GSM8K and MMLU-STEM in Table 1, with 3B and 7B models achieving $50\%$ and $61.7\%$ absolute improvement. The success on Sudoku can be attributed to two factors. First, Sudoku has a unique property where verification is is simpler than generation. While solving requires sophisticated search and constraint satisfaction, verification only requires checking if each row, column and 2x2 sub-grid contains unique digits. This asymmetry makes it ideal for feedback training, as the task is very simple and straightforward. Second, errors in Sudoku are easy to detect in the middle of the solution, and feedback model can provide feedback that helps solver model to correct its mistakes. This is different from MMLU-STEM where the error is more intrinsic and nuanced. The performance difference between model sizes is also revealing. The 1.5B model shows virtually no improvement ($1.7\%$ to $1.8\%$), suggesting a minimum capability threshold for constraint tracking. However, once this threshold is crossed, as with the 3B model, we see massive gains. The 7B model's near-perfect $96.8\%$ accuracy after FRL training indicates that larger models have latent constraint-satisfaction capabilities that can be unlocked through proper feedback, even when their initial performance is poor. On the other hand, all prompting-based approaches did not work at all. They are all showing overconfident issue and therefore no correction would be triggered. If the feedback model is not trained and encounters something unusual, it is more likely to be overconfident and lead to false negative errors.

## 6 Discussion

### 6.1 Reward Choices

Table 5: Reward Ablation Study on GSM8K

| Model Size | Refinement Only | Refinement + Verification (No Reward Shaping) | Refinement + Verification |
|---|---|---|---|
| 3B | 74.8 | 80.7 | **83.1** |
| 7B | 85.1 | 90.5 | **91.8** |

We evaluate the contribution of different reward components to understand their impact on FRL's performance in Table 5. We test three configurations: refinement reward only, refinement with verification reward (no penalty), and refinement with verification reward including false positive penalty. The penalty specifically assigns -1 reward when the feedback incorrectly identifies a correct initial response as incorrect, encouraging more conservative verification.

The results validate our reward design choice. Using refinement reward alone improves the solver model and achieves the highest incorrect-to-correct rate by above $10\%$. However, it lacks the ability to preserve correct ones, frequently giving negative feedback to originally correct initial responses, leading to changing them from correct to incorrect. Adding a verification reward without penalty improves performance upon the refinement reward only setting, as the model now has to optimize for verification accuracy. However, we still observe numerous false positives where initial responses are wrongly flagged and corrupted. Therefore, we introduce the penalty to the verification reward. This reward successfully achieved our goal of lowering the rate of going from correct to incorrect. This reward also achieves the highest accuracy for the two model sizes in the GSM8K setting. The penalty prevents the feedback model from making unnecessary modifications to the correct initial responses while maintaining the ability to improve the incorrect ones. This $2.4\%$ and $1.3\%$ improvement over the no-penalty configuration demonstrates that preventing over-correction is crucial for effective self-correction. The result from the reward ablation study shows that both components, learning to generate helpful feedback and learning when not to intervene, are essential for FRL's success.

## 6.2 Comparison With Direct Solver Model Training

To understand FRL's effectiveness, we compare it against directly training the solver model with GRPO on the same tasks in Table 4. Here we used original questions as input, and answer correctness as reward. This comparison helps establish whether training a separate feedback model offers advantages over simply improving the solver itself. We also evaluate generalization using GSM-Symbolic Mirzadeh et al. (2025), a dataset that tests robustness by permuting variable names and numbers from GSM8K problems. Prior work has shown that most LLMs experience performance degradation on this variant compared to the original GSM8K. Note here we train the models on original GSM8K problems, and test their generalization using GSM-Symbolic.

GRPO-trained solver substantially outperform prompt-based baselines, confirming that reinforcement learning can improve reasoning capabilities more than prompting. However, these gains vary significantly by task and model size. On GSM8K, smaller models see modest improvement, while larger models like 7B benefit more. Comparing this with FRL, we can see GRPO-solvers show accuracy consistently lower than FRL, particularly on smaller models where the gap reaches $10\%$. This could be due to smaller models being harder to train effectively. Even after GRPO training, they still frequently make errors that are difficult to self-correct without external feedback.

The generalization test on GSM-Symbolic reveals interesting patterns. FRL outperforms direct GRPO training by substantial margins for the 1.5B ($12.6\%$) and 3B ($7.7\%$) models. However, the 7B GRPO-trained solver slightly surpasses FRL, achieving $78.2\%$ versus $77.0\%$. This suggests that larger models can better internalize robust problem-solving strategies through direct training, while smaller models benefit more from the structured feedback approach of FRL.

These results demonstrate that smaller models like 1.5B or 3B benefit more from FRL's feedback-based design. The small models struggle with new questions even when trained directly as a solver. But when trained as a feedback model with FRL, these models overcome their inherent limitations. While larger models like 7B achieve competitive performance through direct training, this approach requires accessing model weights, not something we can do with proprietary APIs. Notably, FRL remains competitive even against white-box training methods, validating its practical value and generalization ability for real-world black-box optimization scenarios.

## 7 Conclusion

In this paper we present Feedback Reinforcement Learning (FRL), a method to optimize black-box LLMs by training a separate model to generate corrective feedback. Our experiments across diverse reasoning tasks show FRL significantly boosting accuracy, being generalizable, and outperforming both existing self-correction methods and oracle-based baselines. FRL therefore provides a practical and effective strategy for optimizing proprietary models without requiring access to their weights.

## LLM USAGE STATEMENT

We used an LLM as a grammar checker in this paper.

## REPRODUCIBILITY STATEMENT

We provide detailed methodology of Feedback Reinforcement Learning in Section 3. We have our Experiment setup in Section 5, and Appendix A. We submit our code in the Supplementary Materials and plan to open-source it in a public GitHub repository with full documentation when the paper is published. All pretrained LLMs used were obtained from HuggingFace.

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

# A IMPLEMENTATION

## A.1 TRAINING CONFIGURATION

**Hardware Setup:** All experiments were conducted on the HPC cluster equipped with 4x NVIDIA H200 GPUs (141GB memory each), approximately 600GB total system memory, and CUDA version 13.0.

**Model Training Parameters:** We used base models from the Qwen2.5 family (Qwen et al., 2025) with 1.5B, 3B, and 7B parameters. Training was performed using GRPO (Group Relative Policy Optimization) with the following configuration:

- Learning rate: $1 \times 10^{-6}$
- Per-device batch size: 36
- Gradient accumulation steps: 4
- Effective batch size: 432
- Training epochs: 1
- Weight decay: 0.01
- Warmup ratio: 0.05
- Maximum sequence length: 2048
- Maximum new tokens: 512

**Generation Parameters:** During training, we used the following generation settings:

- Number of generations per step: 24
- Temperature: 0.7
- Top-p sampling: 1.0

**Optimization Details:** We employed Paged AdamW 8-bit optimizer with Adam $\beta_1 = 0.9$, $\beta_2 = 0.999$, and $\epsilon = 1 \times 10^{-8}$. Training used bfloat16 precision with gradient checkpointing enabled. Following Liu et al. (2025), we set the KL penalty to 0.

**Data Generation:** For each dataset, we generated 20,000 question-answer pairs through iterative sampling to maintain a balanced 1:1 ratio of positive (correct initial responses) to negative (incorrect initial responses) examples.

**Reward Configuration:** The final reward combined two components with equal weighting:

- Verification reward: $R_{\text{verify}} = +1$ for correct identification, $-1$ for false positive (penalty), 0 for false negative
- Refinement reward: $R_{\text{refine}} = +1$ for incorrect→correct transitions, 0 otherwise
- Final reward: $R_{\text{final}} = R_{\text{verify}} + R_{\text{refine}}$

**Evaluation Setup:** During evaluation, we used 5-shot Chain-of-Thought prompting for initial problem solving and 2-shot Chain-of-Thought prompting for feedback generation and refinement due to context length constraints. Answers were extracted using boxed format for GSM8K and MMLU-STEM, and grid format for Sudoku puzzles. For baseline comparisons, Self-Refine and Reflexion were limited to a maximum of 5 iterations.

# B PROMPTS

## B.1 GSM8K

### B.1.1 INFERENCE PROMPT

Instruction: Please reason step by step, and put your final answer within $\square$.

Problem: Natalia sold clips to 48 of her friends in April, and then she sold half as many clips in May. How many clips did Natalia sell altogether in April and May?

Solution: To find the total number of clips Natalia sold:

Step 1: Clips sold in April: 48 clips

Step 2: Clips sold in May: half of April's sales = $48 \div 2 = 24$ clips

Step 3: Total clips sold: $48 + 24 = 72$ clips

Therefore, Natalia sold $\boxed{72}$ clips altogether in April and May.

Problem: Weng earns $12 an hour for babysitting. Yesterday, she just did 50 minutes of babysitting. How much did she earn?

Solution: To calculate Weng's earnings:

Step 1: Hourly rate: $12 per hour

Step 2: Time worked: 50 minutes = $\frac{50}{60}$ hours = $\frac{5}{6}$ hours

Step 3: Earnings: $12 \times \frac{5}{6} = \$12 \times 5 \div 6 = \$60 \div 6 = \$10$

Therefore, Weng earned $\boxed{10}$ dollars.

Problem: Betty is saving money for a new wallet which costs $100. Betty has only half of the money she needs. Her parents decided to give her $15 for that purpose, and her grandparents twice as much as her parents. How much more money does Betty need to buy the wallet?

Solution: To find how much more money Betty needs:

Step 1: Wallet cost: $100

Step 2: Betty's current savings: half of $100 = $50

Step 3: Money from parents: $15

Step 4: Money from grandparents: $2 \times \$15 = \$30$

Step 5: Total money Betty has: $\$50 + \$15 + \$30 = \$95$

Step 6: Money still needed: $\$100 - \$95 = \$5$

Therefore, Betty needs $\boxed{5}$ more dollars to buy the wallet.

Problem: Janet's ducks lay 16 eggs per day. She eats three for breakfast every morning and bakes muffins for her friends every day with four. She sells the remainder at the farmers' market daily for $2 per fresh duck egg. How much in dollars does she make every day at the farmers' market?

Solution: To determine how much Janet makes every day at the farmers' market:

Step 1: Total eggs laid per day: 16 eggs

Step 2: Eggs eaten for breakfast: 3 eggs

Step 3: Eggs used for muffins: 4 eggs

Step 4: Eggs left for sale: $16 - 3 - 4 = 9$ eggs

Step 5: Revenue from selling eggs: 9 eggs $\times$ $2 per egg = $18

Therefore, Janet makes $\boxed{18}$ dollars every day at the farmers' market.

Problem: A robe takes 2 bolts of blue fiber and half that much white fiber. How many bolts in total does it take?

Solution: To find the total number of bolts needed:

Step 1: Blue fiber required: 2 bolts

Step 2: White fiber required: half of the blue fiber = $2 \div 2 = 1$ bolt

Step 3: Total bolts needed: 2 bolts (blue) $+1$ bolt (white) $= 3$ bolts

Therefore, it takes $\boxed{3}$ bolts in total to make a robe.

Problem: question

Solution:

### B.1.2   FEEDBACK PROMPT

Instructions:

1. Review the solution step-by-step
2. Check if the solution correctly addresses the problem
3. Verify all calculations and logical steps
4. After completing your analysis, provide your conclusion in final assessment.
5. You have to include either "there is no error" or "there is at least one error" in the conclusion.

Problem: Janet's ducks lay 16 eggs per day. She eats three for breakfast every morning and bakes muffins for her friends every day with four. She sells the remainder at the farmers' market daily for $2 per fresh duck egg. How much in dollars does she make every day at the farmers' market?

Solution: To determine how much Janet makes:

Step 1: Total eggs laid per day: 16 eggs

Step 2: Eggs eaten for breakfast: 3 eggs

Step 3: Eggs used for muffins: 4 eggs

Step 4: Eggs left for sale: $16 - 3 - 4 = 9$ eggs

Step 5: Revenue: 9 eggs $\times$ \$2 $=$ \$18

Therefore, Janet makes $\boxed{18}$ every day.

Analysis:

Step 1 Verification: The problem states Janet's ducks lay 16 eggs per day. This is correctly identified.

Step 2 Verification: The problem states she eats three eggs for breakfast. This is correctly recorded as 3 eggs.

Step 3 Verification: The problem states she uses four eggs for muffins. This is correctly recorded as 4 eggs.

Step 4 Verification: The calculation $16 - 3 - 4 = 9$ is arithmetically correct.

Step 5 Verification: The calculation $9 \times \$2 = \$18$ is arithmetically correct.

Final Assessment:

There is no error.

Problem: A robe takes 2 bolts of blue fiber and half that much white fiber. How many bolts in total does it take?

Solution: To find the total bolts needed:

Step 1: Blue fiber required: 2 bolts

Step 2: White fiber required: half of blue $= 2/2 = 1$ bolt

Step 3: Total bolts: $2 + 1 = 4$ bolts

The total is $\boxed{4}$ bolts.

Analysis:

Step 1 Verification: The problem states 2 bolts of blue fiber. This is correctly identified.

Step 2 Verification: "Half that much white fiber" means half of 2 bolts. The calculation $2/2 = 1$ is arithmetically correct.

Step 3 Verification: The addition shows $2 + 1 = 4$, but this is incorrect arithmetic. $2 + 1 = 3$, not 4.

Final Assessment:

There is at least one error: The addition in Step 3 incorrectly states $2 + 1 = 4$.

Problem: {question}

Solution: {initial_response}

Analysis:

### B.1.3  REFINE PROMPT

Original Problem: Janet's ducks lay 16 eggs per day. She eats three for breakfast every morning and bakes muffins for her friends every day with four. She sells the remainder at the farmers' market daily for $2 per fresh duck egg. How much in dollars does she make every day at the farmers' market?

Previous Solution: To determine how much Janet makes:

Step 1: Total eggs laid per day: 16 eggs

Step 2: Eggs eaten for breakfast: 3 eggs

Step 3: Eggs used for muffins: 4 eggs

Step 4: Eggs left for sale: $16 - 3 - 4 = 10$ eggs

Step 5: Revenue: 10 eggs $\times$ \$2 $=$ \$20

Therefore, Janet makes $\boxed{20}$ dollars every day.

Feedback Received: Step 1 Verification: Correctly identifies 16 eggs laid per day. Step 2 Verification: Correctly identifies 3 eggs for breakfast. Step 3 Verification: Correctly identifies 4 eggs for muffins. Step 4 Verification: The subtraction is incorrect. $16 - 3 - 4 = 9$, not 10. Step 5 Verification: Cannot verify due to error in Step 4. Final Assessment: There is at least one error: The subtraction in Step 4 is incorrect.

Revised Solution: To determine how much Janet makes:

Step 1: Total eggs laid per day: 16 eggs

Step 2: Eggs eaten for breakfast: 3 eggs

Step 3: Eggs used for muffins: 4 eggs

Step 4: Eggs left for sale: $16 - 3 - 4 = 9$ eggs

Step 5: Revenue: 9 eggs $\times$ \$2 $=$ \$18

Therefore, Janet makes $\boxed{18}$ dollars every day.

Original Problem: A robe takes 2 bolts of blue fiber and half that much white fiber. How many bolts in total does it take?

Previous Solution: To find the total bolts needed:

Step 1: Blue fiber required: 2 bolts

Step 2: White fiber required: half of blue = $2/2 = 1$ bolt

Step 3: Total bolts: $2 + 1 = 4$ bolts

The total is $\boxed{4}$ bolts.

Feedback Received: Step 1 Verification: Correctly identifies 2 bolts of blue fiber. Step 2 Verification: Correctly calculates half of 2 as 1 bolt. Step 3 Verification: The addition is incorrect. $2 + 1 = 3$, not 4. Final Assessment: There is at least one error: The addition in Step 3 incorrectly states $2 + 1 = 4$.

Revised Solution: To find the total bolts needed:

Step 1: Blue fiber required: 2 bolts

Step 2: White fiber required: half of blue = $2/2 = 1$ bolt

Step 3: Total bolts: $2 + 1 = 3$ bolts

The total is $\boxed{3}$ bolts.

Original Problem: {question}

Previous Solution: {initial_response}

Feedback Received: {feedback}

Revised Solution:

## B.2   MMLU-STEM

### B.2.1   INFERENCE PROMPT

Instruction: Please reason step by step, and put your final answer within $\boxed{}$.

Problem: Statement 1— RoBERTa pretrains on a corpus that is approximate 10x larger than the corpus BERT pretrained on. Statement 2— ResNeXts in 2018 usually used tanh activation functions. Option 0: True, True Option 1: False, False Option 2: True, False Option 3: False, True

Solution: To evaluate both statements:

Step 1: Statement 1 - RoBERTa vs BERT corpus size: - BERT pretrained on BookCorpus (800M words) + English Wikipedia (2,500M words) $\approx$ 3.3B words - RoBERTa pretrained on BookCorpus + Wikipedia + CC-News + OpenWebText + Stories $\approx$ 160GB of text - RoBERTa's corpus is indeed approximately 10x larger than BERT's. Statement 1 is TRUE.

Step 2: Statement 2 - ResNeXt activation functions: - ResNeXt, introduced in 2017 and widely used in 2018, is based on ResNet architecture - Like ResNet, ResNeXt uses ReLU (Rectified Linear Unit) activation functions, not tanh - Tanh was more common in older architectures; modern CNNs predominantly use ReLU. Statement 2 is FALSE.

Step 3: Combine evaluations: True, False corresponds to Option 2.

Therefore, the answer is $\boxed{2}$.

Problem: The Barkhausen criterion for an oscillator Option 0: Loop gain should be unity Option 1: Loop gain should be less than unity Option 2: The phase of a feedback signal with respect to input should be 0 or 360 Option 3: Both A and C

Solution: To identify the Barkhausen criterion requirements:

Step 1: Understand oscillator requirements - For sustained oscillations, a circuit needs specific conditions for the feedback loop.

Step 2: Loop gain condition - The magnitude of the loop gain (product of forward gain and feedback factor) must equal 1 (unity) for sustained oscillations. This matches Option 0.

Step 3: Phase condition - The total phase shift around the loop must be 0 or 360 (or integer multiples of 360) for positive feedback. This matches Option 2.

Step 4: Complete Barkhausen criterion - Both conditions must be satisfied simultaneously: - |Loop gain| = 1 (unity) - Phase shift = 0 or 360

Step 5: Match to options - Both Option 0 and Option 2 are required, which corresponds to Option 3.

Therefore, the answer is $\boxed{3}$.

Problem: The quantum efficiency of a photon detector is 0.1. If 100 photons are sent into the detector, one after the other, the detector will detect photons Option 0: an average of 10 times, with an rms deviation of about 4 Option 1: an average of 10 times, with an rms deviation of about 3 Option 2: an average of 10 times, with an rms deviation of about 1 Option 3: an average of 10 times, with an rms deviation of about 0.1

Solution: To find the detection statistics:

Step 1: Calculate expected detections - With quantum efficiency $\eta = 0.1$ and $n = 100$ photons: - Expected number of detections = $\eta \times n = 0.1 \times 100 = 10$

Step 2: Determine the statistical distribution - Photon detection follows binomial distribution with: - Number of trials: $n = 100$ - Probability of success: $p = 0.1$

Step 3: Calculate standard deviation for binomial distribution: - Standard deviation $\sigma = \sqrt{n \times p \times (1-p)}$ - $\sigma = \sqrt{100 \times 0.1 \times 0.9} = \sqrt{9} = 3$

Step 4: RMS deviation equals standard deviation for this process = 3

Step 5: Match to options - Average of 10 with RMS deviation of 3 corresponds to Option 1.

Therefore, the answer is $\boxed{1}$.

Problem: A gas phase is generally absent from which of the following biogeochemical cycles? Option 0: Water Option 1: Carbon Option 2: Sulfur Option 3: Phosphorus

Solution: To determine which biogeochemical cycle generally lacks a gas phase:

Step 1: Water cycle - Includes water vapor ($H_2O$ gas) in the atmosphere through evaporation and condensation. Gas phase is present.

Step 2: Carbon cycle - Includes carbon dioxide ($CO_2$) and methane ($CH_4$) gases exchanged between atmosphere, organisms, and oceans. Gas phase is present.

Step 3: Sulfur cycle - Includes sulfur dioxide ($SO_2$) and hydrogen sulfide ($H_2S$) gases from volcanic emissions and decomposition. Gas phase is present.

Step 4: Phosphorus cycle - Phosphorus primarily cycles through rocks, soil, water, and organisms in solid or dissolved forms. Under normal Earth conditions, phosphorus does not form stable gaseous compounds. Gas phase is generally absent.

Therefore, the answer is $\boxed{3}$.

Problem: What is the units digit in the standard decimal expansion of the number $7^{25}$? Option 0: 1 Option 1: 3 Option 2: 5 Option 3: 7

Solution: To find the units digit of $7^{25}$:

Step 1: Calculate the first few powers of 7 and identify units digits: - $7^1 = 7$ (units digit: 7) - $7^2 = 49$ (units digit: 9) - $7^3 = 343$ (units digit: 3) - $7^4 = 2401$ (units digit: 1) - $7^5 = 16807$ (units digit: 7)

Step 2: Identify the pattern: The units digits follow the pattern $[7, 9, 3, 1]$ and repeat every 4 powers.

Step 3: Determine position in cycle: $25 \div 4 = 6$ with remainder 1

Step 4: Apply the remainder: Since the remainder is 1, $7^{25}$ has the same units digit as $7^1$, which is 7.

Step 5: Match to option: Units digit 7 corresponds to Option 3.

Therefore, the answer is $\boxed{3}$.

Problem: {question}

Solution:

### B.2.2 FEEDBACK PROMPT

Instructions:

1. Review the solution step-by-step
2. Check if the solution correctly addresses the problem
3. Verify all calculations and logical steps
4. After completing your analysis, provide your conclusion in final assessment.
5. You have to include either "there is no error" or "there is at least one error" in the conclusion.

Problem: A gas phase is generally absent from which of the following biogeochemical cycles? Option 0: Water Option 1: Carbon Option 2: Sulfur Option 3: Phosphorus

Solution: Analyzing each cycle:

Step 1: Water cycle has water vapor in atmosphere (gas present)

Step 2: Carbon cycle has $CO_2$ exchanges (gas present)

Step 3: Sulfur cycle has $SO_2$ emissions (gas present)

Step 4: Phosphorus cycle has no significant gas phase

Step 5: The answer is Option $\boxed{3}$.

Analysis:

Step 1 Verification: Water vapor is indeed a gas phase in the water cycle. This is correct.

Step 2 Verification: $CO_2$ is a gas in the carbon cycle. This is correct.

Step 3 Verification: $SO_2$ and other sulfur gases exist in the sulfur cycle. This is correct.

Step 4 Verification: Phosphorus does not have a significant gas phase under normal conditions. This is correct.

Step 5 Verification: Option 3 (Phosphorus) correctly corresponds to the cycle without a gas phase.

Final Assessment:

There is no error.

Problem: What is the units digit in the standard decimal expansion of the number $7^{25}$? Option 0: 1 Option 1: 3 Option 2: 5 Option 3: 7

Solution: To find the units digit of $7^{25}$:

Step 1: Calculate pattern - $7^1 = 7$, $7^2 = 49$, $7^3 = 343$, $7^4 = 2401$

Step 2: Units digits form pattern: $[7, 9, 3, 1]$

Step 3: Pattern repeats every 4 powers

Step 4: $25 \div 4 = 6$ remainder 1

Step 5: Remainder 1 corresponds to first position, units digit 7

Step 6: The answer is Option $\boxed{1}$.

Analysis:

Step 1 Verification: The calculations are correct with units digits $7, 9, 3, 1$.

Step 2 Verification: The pattern $[7, 9, 3, 1]$ is correctly identified.

Step 3 Verification: The pattern does repeat every 4 powers.

Step 4 Verification: $25 \div 4 = 6$ remainder $1$ is arithmetically correct.

Step 5 Verification: Remainder $1$ does correspond to units digit $7$.

Step 6 Verification: Units digit $7$ corresponds to Option 3, not Option 1.

Final Assessment:

There is at least one error: Option 1 is incorrectly selected when units digit 7 corresponds to Option 3.

Problem: {question}

Solution: {initial_response}

Analysis:

### B.2.3 REFINE PROMPT

Original Problem: A gas phase is generally absent from which of the following biogeochemical cycles? Option 0: Water Option 1: Carbon Option 2: Sulfur Option 3: Phosphorus

Previous Solution: Analyzing each cycle:

Step 1: Water has water vapor (gas present)

Step 2: Carbon has $CO_2$ (gas present)

Step 3: Sulfur has $SO_2$ (gas present)

Step 4: Phosphorus has no gas phase

Step 5: The answer is $\boxed{2}$.

Feedback Received: Step 1 Verification: Water vapor is a gas - correct. Step 2 Verification: $CO_2$ is a gas - correct. Step 3 Verification: $SO_2$ is a gas - correct. Step 4 Verification: Phosphorus lacks gas phase - correct. Step 5 Verification: Phosphorus is Option 3, not Option 2. Final Assessment: There is at least one error: Option 2 is incorrectly selected when the answer should be Option 3.

Revised Solution: Analyzing each cycle:

Step 1: Water has water vapor (gas present)

Step 2: Carbon has $CO_2$ (gas present)

Step 3: Sulfur has $SO_2$ (gas present)

Step 4: Phosphorus has no gas phase

Step 5: The answer is $\boxed{3}$.

Original Problem: What is the units digit in the standard decimal expansion of the number $7^{25}$? Option 0: 1 Option 1: 3 Option 2: 5 Option 3: 7

Previous Solution: To find units digit of $7^{25}$:

Step 1: Powers give units digits: 7, 9, 3, 1

Step 2: Pattern repeats every 4

Step 3: $25 \div 4 = 5$ remainder 5

Step 4: Remainder 5 wraps to position 1, units digit 7

Step 5: The answer is $\boxed{1}$.

Feedback Received: Step 1 Verification: Pattern 7, 9, 3, 1 is correct. Step 2 Verification: Cycle length 4 is correct. Step 3 Verification: $25 \div 4 = 6$ remainder 1, not 5 remainder 5. Step 4 Verification: Cannot verify due to error in Step 3. Step 5 Verification: Cannot verify due to error in Step 3. Final Assessment: There is at least one error: The division $25 \div 4$ is incorrectly calculated.

Revised Solution: To find units digit of $7^{25}$:

Step 1: Powers give units digits: 7, 9, 3, 1

Step 2: Pattern repeats every 4

Step 3: $25 \div 4 = 6$ remainder 1

Step 4: Remainder 1 means position 1, units digit 7

Step 5: The answer is $\boxed{3}$.

Original Problem: {question}

Previous Solution: {initial_response}

Feedback Received: {feedback}

Revised Solution:

### B.3   SUDOKU PUZZLE

#### B.3.1   INFERENCE PROMPT

Instruction: Solve this 4x4 sudoku step by step where each row, column, and 2x2 box must contain 1-4 exactly once, then put your final answer grid inside verbatim blocks.

Problem:

```
1 2 3 4
4 . 2 .
. 1 . 3
3 . 1 .
```

Solution: Step 1: Analyze row 2

Row 2: `4 .   2   .`

Missing numbers: 1, 3

- Column 2: Already has 2, 1 in other rows, so position (2,2) must be 3
- Column 4: Must be 1

Grid after step 1:

```
1 2 3 4
4 3 2 1
. 1 . 3
3 . 1 .
```

Step 2: Analyze column 1

Column 1: `1 4 .   3`

Missing number: 2

- Position (3,1) must be 2

Grid after step 2:

```
1 2 3 4
4 3 2 1
```

```
2 1 . 3
3 . 1 .
```

Step 3: Analyze row 3

Row 3: `2 1 .  3`

Missing number: 4

- Position (3,3) must be 4

Grid after step 3:

```
1 2 3 4
4 3 2 1
2 1 4 3
3 . 1 .
```

Step 4: Analyze column 2

Column 2: `2 3 1 .`

Missing number: 4

- Position (4,2) must be 4

Grid after step 4:

```
1 2 3 4
4 3 2 1
2 1 4 3
3 4 1 .
```

Step 5: Analyze row 4

Row 4: `3 4 1 .`

Missing number: 2

- Position (4,4) must be 2

Final Answer:

```
1 2 3 4
4 3 2 1
2 1 4 3
3 4 1 2
```

Problem:

```
1 2 3 4
4 . 2 .
. 1 4 3
3 4 1 .
```

Solution: Step 1: Analyze row 2

Row 2: `4 .  2 .`

Missing numbers: 1, 3

- Column 2: Already has 2, 1, 4 in other rows, so position (2,2) must be 3
- Column 4: Must be 1

Grid after step 1:

```
1 2 3 4
4 3 2 1
. 1 4 3
3 4 1 .
```

Step 2: Analyze column 1

Column 1: `1  4  .   3`

Missing number: 2

- Position (3,1) must be 2

Grid after step 2:

```
1 2 3 4
4 3 2 1
2 1 4 3
3 4 1 .
```

Step 3: Analyze row 4

Row 4: `3 4 1 .`

Missing number: 2

- Position (4,4) must be 2

Final Answer:

```
1 2 3 4
4 3 2 1
2 1 4 3
3 4 1 2
```

Problem:

```
1 2 3 4
4 . 2 1
2 1 4 3
3 4 1 .
```

Solution: Step 1: Analyze row 2

Row 2: `4  .   2  1`

Missing number: 3

- Position (2,2) must be 3

Grid after step 1:

```
1 2 3 4
4 3 2 1
2 1 4 3
3 4 1 .
```

Step 2: Analyze row 4

Row 4: `3 4 1 .`

Missing number: 2

- Position (4,4) must be 2

Final Answer:

```
1 2 3 4
4 3 2 1
2 1 4 3
3 4 1 2
```

Problem:

```
{question}
```

Solution:

### B.3.2 FEEDBACK PROMPT

Instructions:

1. Review the solution step-by-step
2. Check if the solution correctly addresses the problem
3. Verify all calculations and logical steps
4. After completing your analysis, provide your conclusion in final assessment
5. You have to include either "there is no error" or "there is at least one error" in the conclusion

Problem:

```
1 2 3 4
4 . 2 .
. 1 . 3
3 . 1 .
```

Solution:

Step 1: Analyze row 2

Row 2: `4 .   2 .`

Missing numbers: 1, 3

- Column 2: Already has 2, 1 in other rows, so position (2,2) must be 3
- Column 4: Must be 1

Grid after step 1:

```
1 2 3 4
4 3 2 1
. 1 . 3
3 . 1 .
```

Step 2: Analyze column 1

Column 1: `1 4 .   3`

Missing number: 2

- Position (3,1) must be 2

Grid after step 2:

```
1 2 3 4
4 3 2 1
2 1 . 3
3 . 1 .
```

Step 3: Analyze row 3

Row 3: `2 1 .  3`

Missing number: 4

- Position (3,3) must be 4

Grid after step 3:

```
1 2 3 4
4 3 2 1
2 1 4 3
3 . 1 .
```

Step 4: Analyze column 2

Column 2: `2 3 1 .`

Missing number: 4

- Position (4,2) must be 4

Grid after step 4:

```
1 2 3 4
4 3 2 1
2 1 4 3
3 4 1 .
```

Step 5: Analyze row 4

Row 4: `3 4 1 .`

Missing number: 3

- Position (4,4) must be 3

Final Answer:

```
1 2 3 4
4 3 2 1
2 1 4 3
3 4 1 3
```

Analysis:

Step 1 Verification: Row 2 has 4, *, 2, *. Missing numbers are indeed 1 and 3. Column 2 check: row 1 has 2, row 3 has 1, so position (2,2) can be 3. Column 4 would then be 1. This is logically correct.

Step 2 Verification: Column 1 has 1, 4, _, 3. The missing number is 2, so position (3,1) = 2 is correct.

Step 3 Verification: Row 3 now has 2, 1, _, 3. The missing number is 4, so position (3,3) = 4 is correct.

Step 4 Verification: Column 2 has 2, 3, 1, _. The missing number is 4, so position (4,2) = 4 is correct.

Step 5 Verification: Row 4 has 3, 4, 1, _. The numbers 1, 3, 4 are present. The missing number should be 2, not 3. The solution incorrectly states the missing number is 3, which would create a duplicate 3 in row 4.

Final Assessment: There is at least one error. Step 5 incorrectly identifies the missing number as 3 instead of 2, resulting in an invalid Sudoku solution with duplicate 3s in row 4.

Problem:

```
1 2 3 4
4 . 2 .
. 1 4 3
3 4 1 .
```

Solution:

Step 1: Analyze row 2

Row 2: `4 . 2 .`

Missing numbers: 1, 3

- Column 2: Already has 2, 1, 4 in other rows, so position (2,2) must be 3
- Column 4: Must be 1

Grid after step 1:

```
1 2 3 4
4 3 2 1
. 1 4 3
3 4 1 .
```

Step 2: Analyze column 1

Column 1: `1 4 . 3`

Missing number: 2

- Position (3,1) must be 2

Grid after step 2:

```
1 2 3 4
4 3 2 1
2 1 4 3
3 4 1 .
```

Step 3: Analyze row 4

Row 4: `3 4 1 .`

Missing number: 2

- Position (4,4) must be 2

Final Answer:

```
1 2 3 4
4 3 2 1
2 1 4 3
3 4 1 2
```

Analysis:

Step 1 Verification: Row 2 has 4, *, 2, *. Missing numbers are 1 and 3. Column 2 check: row 1 has 2, row 3 has 1, row 4 has 4, so position (2,2) must be 3. Column 4 would then be 1. This is correct.

Step 2 Verification: Column 1 has 1, 4, _, 3. The missing number is 2, so position (3,1) = 2 is correct.

Step 3 Verification: Row 4 has 3, 4, 1, _. The numbers 1, 3, 4 are present. The missing number is 2, so position (4,4) = 2 is correct.

Final grid check:

- All rows contain 1, 2, 3, 4 exactly once

- All columns contain 1, 2, 3, 4 exactly once
- The solution matches the original given numbers

Final Assessment: There is no error.

Problem:

{question}

Solution: initial_response

Analysis:

### B.3.3   REFINE PROMPT

Original Problem

```
1 2 3 4
4 . 2 .
. 1 . 3
3 . 1 .
```

Previous Solution:

Step 1: Analyze row 2

Row 2: `4 .   2 .`

Missing numbers: 1, 3

- Column 2: Already has 2, 1 in other rows, so position (2,2) must be 3
- Column 4: Must be 1

Grid after step 1:

```
1 2 3 4
4 3 2 1
. 1 . 3
3 . 1 .
```

Step 2: Analyze column 1

Column 1: `1 4 .   3`

Missing number: 2

- Position (3,1) must be 2

Grid after step 2:

```
1 2 3 4
4 3 2 1
2 1 . 3
3 . 1 .
```

Step 3: Analyze row 3

Row 3: `2 1 .   3`

Missing number: 4

- Position (3,3) must be 4

Grid after step 3:

```
1 2 3 4
4 3 2 1
2 1 4 3
3 . 1 .
```

Step 4: Analyze column 2

Column 2: `2  3  1  .`

Missing number: 4

- Position (4,2) must be 4

Grid after step 4:

```
1 2 3 4
4 3 2 1
2 1 4 3
3 4 1 .
```

Step 5: Analyze row 4

Row 4: `3  4  1  .`

Missing number: 3

- Position (4,4) must be 3

Final Answer:

```
1 2 3 4
4 3 2 1
2 1 4 3
3 4 1 3
```

Feedback Received:

Step 1 Verification: Row 2 has 4, ␣, 2, ␣. Missing numbers are 1 and 3. Column 2 check: row 1 has 2, row 3 has 1, so position (2,2) can be 3. Column 4 would be 1. This is correct.

Step 2 Verification: Column 1 has 1, 4, ␣, 3. The missing number is 2, so position (3,1) = 2 is correct.

Step 3 Verification: Row 3 now has 2, 1, ␣, 3. The missing number is 4, so position (3,3) = 4 is correct.

Step 4 Verification: Column 2 has 2, 3, 1, ␣. The missing number is 4, so position (4,2) = 4 is correct.

Step 5 Verification: Row 4 has 3, 4, 1, ␣. The solution states the missing number is 3, but 3 is already present in the row. The missing number should be 2.

Final Assessment: There is at least one error: Step 5 incorrectly identifies the missing number as 3 instead of 2.

Revised Solution:

Step 1: Analyze row 2

Row 2: `4  .    2  .`

Missing numbers: 1, 3

- Column 2: Already has 2, 1 in other rows, so position (2,2) must be 3
- Column 4: Must be 1

Grid after step 1:

```
1 2 3 4
```

```
4 3 2 1
. 1 . 3
3 . 1 .
```

Step 2: Analyze column 1

Column 1: `1  4  .    3`

Missing number: 2

- Position (3,1) must be 2

Grid after step 2:

```
1 2 3 4
4 3 2 1
2 1 . 3
3 . 1 .
```

Step 3: Analyze row 3

Row 3: `2  1  .    3`

Missing number: 4

- Position (3,3) must be 4

Grid after step 3:

```
1 2 3 4
4 3 2 1
2 1 4 3
3 . 1 .
```

Step 4: Analyze column 2

Column 2: `2  3  1  .`

Missing number: 4

- Position (4,2) must be 4

Grid after step 4:

```
1 2 3 4
4 3 2 1
2 1 4 3
3 4 1 .
```

Step 5: Analyze row 4

Row 4: `3  4  1  .`

Missing number: 2

- Position (4,4) must be 2

Final Answer:

```
1 2 3 4
4 3 2 1
2 1 4 3
3 4 1 2
```

Original Problem:

```
1 2 3 4
4 . 2 .
. 1 4 3
3 4 1 .
```

Previous Solution:

Step 1: Analyze row 2

Row 2: `4 .  2 .`

Missing numbers: 1, 3

- Column 2: Already has 2, 1, 4 in other rows, so position (2,2) must be 1
- Column 4: Must be 3

Grid after step 1:

```
1 2 3 4
4 1 2 3
. 1 4 3
3 4 1 .
```

Step 2: Analyze column 1

Column 1: `1 4 .  3`

Missing number: 2

- Position (3,1) must be 2

Grid after step 2:

```
1 2 3 4
4 1 2 3
2 1 4 3
3 4 1 .
```

Step 3: Analyze row 4

Row 4: `3 4 1 .`

Missing number: 2

- Position (4,4) must be 2

Final Answer:

```
1 2 3 4
4 1 2 3
2 1 4 3
3 4 1 2
```

Feedback Received:

Step 1 Verification: Row 2 has 4, ⎵, 2, ⎵. Missing numbers are 1 and 3. Column 2 check: row 1 has 2, row 3 has 1, row 4 has 4. The only missing number in column 2 is 3, not 1. Position (2,2) should be 3.

Step 2 Verification: Based on the error in Step 1, this step cannot be properly verified.

Step 3 Verification: Cannot verify due to propagated errors.

Final Assessment: There is at least one error: Step 1 incorrectly assigns 1 to position (2,2) when it should be 3.

Revised Solution:

Step 1: Analyze row 2

Row 2: `4 . 2 .`

Missing numbers: 1, 3

- Column 2: Already has 2, 1, 4 in other rows, so position (2,2) must be 3
- Column 4: Must be 1

Grid after step 1:

```
1 2 3 4
4 3 2 1
. 1 4 3
3 4 1 .
```

Step 2: Analyze column 1

Column 1: `1 4 . 3`

Missing number: 2

- Position (3,1) must be 2

Grid after step 2:

```
1 2 3 4
4 3 2 1
2 1 4 3
3 4 1 .
```

Step 3: Analyze row 4

Row 4: `3 4 1 .`

Missing number: 2

- Position (4,4) must be 2

Final Answer:

```
1 2 3 4
4 3 2 1
2 1 4 3
3 4 1 2
```

Original Problem:

`{question}`

Previous Solution: initial_response

Feedback Received: feedback

Revised Solution:

## C  COMPLETE GSM8K EXAMPLE

**Input Question:**

Stephen placed an online order for groceries. His final bill came to $40.00. Because this was through a delivery vendor, they tacked on a 25% fee to his final total and charged him $3.00 in delivery fees. Stephen also added a $4.00 tip. After the extra fees, what was the final price of Stephen's groceries?

**Initial Response:**

To find the final price of Stephen's groceries:

Step 1: Final bill: $40.00

Step 2: Delivery fee: $3.00

Step 3: Tip: $4.00

Step 4: Total extra fees: $3.00 (delivery) + $4.00 (tip) = $7.00

Step 5: Total bill with extra fees: $40.00 (final bill) + $7.00 (extra fees) = $47.00

Therefore, the final price of Stephen's groceries was $\boxed{47}$ dollars.

**Feedback:**

Step 1 Verification: The problem states the final bill amount as $40.00. This is correctly identified.

Step 2 Verification: The delivery fee stated is $3.00, which is correctly identified.

Step 3 Verification: The tip stated is $4.00, which is correctly identified.

Step 4 Error: The problem states a 25% service fee on the final bill but does not account for it in the calculation of total extra fees.

Step 5 Error: In Step 5, only the extra fees ($3.00 + $4.00) are added to the bill, without including the 25% service fee.

Final Assessment:

There is at least one error, as the 25% service fee on the final bill and the $3.00 delivery fee (not stated as a service fee) should have been included in the total extra fees.

**Final Response:**

To find the final price of Stephen's groceries:

Step 1: Final bill: $40.00

Step 2: Delivery fee: $3.00

Step 3: Tip: $4.00

Step 4: Calculate the 25% service fee on the final bill: $40.00 \times 0.25 = $10.00

Step 5: Total extra fees: $3.00 (delivery) + $4.00 (tip) + $10.00 (service fee) = $17.00

Step 6: Total bill with extra fees: $40.00 (final bill) + $17.00 (extra fees) = $57.00

Therefore, the final price of Stephen's groceries was $\boxed{57}$ dollars.

Input Question: question

Initial Response: initial_response

Feedback: feedback

Final Response:

