# OpenReview forum: "Improving Black-Box LLMs with Feedback Reinforcement Learning"
_ICLR.cc/2026/Conference — ICLR 2026 Conference Withdrawn Submission_

### Official Review · Reviewer_1mST · 2025-10-30

**Soundness:** 3
**Presentation:** 2
**Contribution:** 2
**Rating:** 4
**Confidence:** 3

**Summary:**

This paper proposes Feedback Reinforcement Learning (FRL) to improve the reasoning ability of black-box large language models (LLMs). Unlike prior approaches that rely on self-reflection or in-context learning (ICL), FRL trains a separate feedback model using GRPO to provide targeted feedback to a solver model. Experiments on GSM8K, MMLU-STEM, and a generated Sudoku benchmark show marginal accuracy gains (on the order of ~10%+) and indicate that the trained feedback model can transfer across model families.

**Strengths:**

- The task is important, and the separation of verification and feedback is clear and well-motivated for black-box scenarios. The reward is structured to help the feedback model distinguish among different response qualities.

- Reported experiments show roughly 10%+ accuracy improvements across the three datasets.

**Weaknesses:**

- The paper would benefit from clearer structure. For example, it lacks an illustrative figure that walks through a single FRL example, and key training details are only briefly described. In addition, the experiments and discussion sections spend substantial space reiterating table numbers rather than providing deeper analysis.

- While the experiments are a good start, they do not fully establish FRL’s feasibility. In particular, out-of-distribution (OOD) reasoning/generalization is not assessed. GSM8K and the generated Sudoku benchmark are relatively simple compared to newer reasoning benchmarks, so further evaluation on more realistic tasks would strengthen the case.

**Questions:**

1. For results of Line 328 vs. Line 335: Intuitively, models of the same size using the same Vanilla Feedback method should yield similar results. Why do we see 62.8 in one case and 85.4 in the other? Is this solely due to the solver/feedback models coming from different families, or are there other factors at play?

2. Inspired by vanilla feedback methods, why not try multiple iterative rounds within FRL? What were the results? Do incremental rounds further improve accuracy? If not, why?

3. Why choose GRPO to train the feedback model? What specific advantages does GRPO (or reinforcement learning more broadly) provide here compared to other optimization methods?

4. Most baselines are training-free (except Table 4, which trains the solver and thus is not black-box). How would a finetuned feedback model (trained via standard supervised finetuning with a similar compute budget to FRL) perform as a baseline or for the ablation study?

---

### Official Review · Reviewer_ALFp · 2025-10-30

**Soundness:** 2
**Presentation:** 2
**Contribution:** 2
**Rating:** 4
**Confidence:** 3

**Summary:**

FRL presents a novel paradigm for improving black-box LLM performance by training a separate feedback model through reinforcement learning. The feedback model learns to identify errors in initial responses and generate corrective feedback, enabling the solver model to refine its outputs without any weight updates.

**Strengths:**

1. Feedback models trained on Qwen successfully guide Llama/Deepseek without ever seeing them during training
2.  Small 3B feedback models effectively improve 7B-14B solvers, offering computational efficiency
3. There are consistent gains across math (GSM8K), knowledge (MMLU), and logic puzzles (Sudoku)

**Weaknesses:**

### 1. Inconsistent baseline performance
In Table 1, there are some interesting results that warrant further investigation. For example, why does Reflexion perform so poorly (19.6) on MMLU-STEM with the 1.5B model while with 3B and 7B models it performs much better (65.3 and 64.9)? Meanwhile, Self-Refine has similar settings as Reflexion, but the performance degradation with the 1.5B model (47.1) is not as severe as Reflexion's.

Also for the FRL results on Sudoku with 3B and 7B models, why is there such a huge performance increase (4.0 initial response vs 54.0 after FRL for 3B, and 35.1 before vs 96.8 after FRL for 7B)? Do the 3B and 7B feedback models provide more actionable feedback than the 1.5B model and the vanilla feedback + oracle verification model? Are there fewer failure modes than in other cases? Does the model action the feedback better? And what's the ratio among all the cases? A more comprehensive analysis is needed.

### 2. Missing evaluation of feedback quality
There is no evaluation of the quality of feedback in FRL. To what extent does FRL perform better than e.g., self-correction feedback and self-verification feedback? Does it generate more accurate feedback? For the training process of FRL, you need to sample from the solver model's output until obtaining a balanced dataset of positive samples and negative samples - what are the computational resources and tokens needed for this? Do you have extra steps for training the quality of the feedback (i.e., how to get more accurate feedback? How to make the feedback easy to follow?)

### 3. Limited task diversity despite acknowledging hard reasoning tasks
The paper gives a comprehensive related work section. In line 96, they mention that self-verification is insufficient for hard reasoning tasks. Hard reasoning tasks could include Reasoning — HotPotQA (Wikipedia multi-hop), Programming — HumanEval, Decision-making — AlfWorld (domestic-robot household tasks), as mentioned in https://arxiv.org/pdf/2412.14959. However, the paper still uses math reasoning tasks mainly to test the model's performance. Math reasoning tasks are designed to be multi-step and are easier to locate faulty reasoning, but the generalization ability to other tasks remains undiscovered.
Missing analysis of feedback application failures
Recent literature also shows that the key bottleneck for self-correction is that models could potentially change their originally correct steps to wrong or correct answers to wrong after the feedback step. The paper did not investigate the model's ability to apply feedback to their answer and did not show the percentage of false negatives (correct→wrong) in their final results, which makes the conclusions less persuasive.

### 4. Missing key RL-based feedback baselines
The paper lacks baselines from Reinforcement Learning for Feedback papers, for example, RL4F (https://arxiv.org/pdf/2305.08844), SCoRe (https://arxiv.org/pdf/2409.12917), CriticGPT (https://arxiv.org/abs/2407.00215), and RLEF (https://arxiv.org/pdf/2410.02089).

**Questions:**

1. What fraction of cases are ✗→✓, ✓→✗, ✓→✓, ✗→✗ after FRL (per model size)? Is there an adherence metric showing the solver actually follows feedback?
2. What’s the observed ✓→✗ rate with/without the false-positive penalty? Confidence intervals?
3. How many feedback rounds are optimal? Does FRL also have the issue with "most of the responses do not go through the self-correction loop"?

---

### Official Review · Reviewer_UjpW · 2025-11-01

**Soundness:** 1
**Presentation:** 2
**Contribution:** 1
**Rating:** 2
**Confidence:** 4

**Summary:**

The paper proposes a framework for enhancing the performance of large language models when direct weight updates are not possible. The method introduces a separate feedback model that is trained via reinforcement learning to identify and correct errors in responses generated by a fixed solver model. During training, the feedback model receives a reward based on two signals: whether it correctly verifies the solver’s initial output and whether the solver’s revised answer, conditioned on the feedback, shows improvement. The approach aims to optimize black-box models indirectly by refining the quality of feedback rather than the solver itself. Experiments on conducted on GSM8K, MMLU-STEM, and Sudoku tasks against training-free baselines such as self-refinement, Reflexion.

**Strengths:**

- The topic is somewhat relevant to the community’s growing interest in improving LLMs through LLM feedback.
- The paper is clearly written and easy to follow, with a straightforward presentation of the proposed method and experiments.

**Weaknesses:**

- Novelty: The core idea is essentially training an LLM as a judge, which has already been explored in prior work such as RL4F [1] and other self-correction works. The paper does not appear to provide substantial new insights beyond existing approaches that train feedback models or critics for LLM refinement.

- Evaluation: The experimental evaluation is weak, relying mainly on simpler benchmarks such as GSM8K and MMLU, and the comparisons are limited to training-free baselines.

[1] Akyürek, Afra Feyza, et al. "RL4F: Generating Natural Language Feedback with Reinforcement Learning for Repairing Model Outputs."

**Questions:**

see above.

---

### Official Review · Reviewer_5c83 · 2025-11-03

**Soundness:** 3
**Presentation:** 3
**Contribution:** 2
**Rating:** 4
**Confidence:** 4

**Summary:**

The paper investigates whether black-box LLMs can be improved by training a separate feedback model via RL, which both verifies the black-box model’s answers and provides feedback on its errors to aid its self-correction. The authors validate their approach with up to 7B feedback models and 14B black-box “solver” models, showing improvements over an untrained-feedback-model baseline on all three studied benchmarks (GSM8k, a STEM subset of MMLU, and 4x4 Sudoku puzzles).

**Strengths:**

- The idea of training a verifier specifically to improve the performance of a black-box model is **interesting and novel**.

- **The breadth of ablations and experimental questions is notable, and results convincing.** The authors measure the efficacy of their Feedback Reinforcement Learning (FRL) method across three model sizes (table 1), generalizability to models unseen during training (table 2), and using smaller models to provide feedback for bigger models (table 3).

- **The major algorithmic design choices are well-supported by experiments**, such as the reward shaping study in section 6.1.

- The **authors took care to provide reasonable and intuitive interpretations of the results.** For example, in section 5.2, where the authors explained FRL’s poor performance on MMLU-STEM with 1.5b model - contrary to FRL’s great performance on GSM8k with 1.5b model - by insightfully pointing out that small models simply lack the requisite domain-specific knowledge for knowledge-intensive tasks and “no amount of feedback can compensate for this knowledge gap.”

**Weaknesses:**

- **The real-world, practical application of the proposed FRL method remains limited in the presence of bigger models with fine-tuning APIs.** According to Table 4, the gap between the “solver” and FRL closes as a) we’re able to perform reinforcement learning directly on the solver model, and b) model size increases. In practice, both cases are in fact true, with major LLM API providers offering a finetuning API, and presumably hosting much larger models than the up to 7b models studied in Section 6.2. Since the paper is predominantly motivated by a potential real-world scenario, it’d be great to see a further discussion on that issue.

- The paper performs all experiments on top of pretrained models with few-shot prompting to “minimize the influence of post-training”. I find this motivation unconvincing, especially because the paper is practically-oriented and most major LLM providers offer API access for post-trained models only. **The paper would benefit by an extension of analyses to reasoning models to better approximate the real-world setting** (e.g GPT-oss-*, R1-Distill-Qwen3-*, or QwQ-32B).

- It’s **unclear if the FRL method scales to the asymmetric case as solver model sizes increase.** In Table 3, the 14B model’s accuracy on GSM8k improves only by 0.7 points absolute (88.9 -> 89.6) with FRL as compared to solver’s initial response. The authors could extend this analysis to other benchmarks to corroborate the trend.

- **The paper lacks a natural baseline in SFT-only training**, applying RL directly on the base model without ablating an SFT-only approach first. Moreover, it’s common practice to sandwich a short round of SFT between pretraining and RL, which is a natural third alternative to ablate.

- Some **minor clarity / ambiguity issues**:
  - Table 1, row 1, column 1 has a typo: “Sover” instead of “Solver”
  - It’s not clear what model family is used for the experiments in Table 3
  - It’s not entirely clear what datasets were used for training. The methods are evaluated on GSM8k, MMLU-STEM, and 4x4 Sudoku - the first has a train set, and the last permits synthetic data generation, but it’s unclear how MMLU-STEM was used for training (if at all), since MMLU lacks a train set.

**Questions:**

See “Weaknesses” section

---

### Note · Authors · 2025-12-11

**Comment:**

We thank all reviewers for their time and thoughtful feedback. We appreciate the recognition of the novelty and effectiveness of our method, and we also acknowledge the areas where further development is needed. Since our initial submission, we have made substantial changes to our method, and in light of the insightful experimental and analytical suggestions provided in the reviews, we have decided to withdraw the current version. We plan to resubmit a significantly strengthened manuscript to a future venue, addressing all issues raised in the reviews.

**Withdrawal Confirmation:**

I have read and agree with the venue's withdrawal policy on behalf of myself and my co-authors.